# *Lactobacillus reuteri* Colonisation of Extremely Preterm Infants in a Randomised Placebo-Controlled Trial

**DOI:** 10.3390/microorganisms9050915

**Published:** 2021-04-24

**Authors:** Johanne E. Spreckels, Erik Wejryd, Giovanna Marchini, Baldvin Jonsson, Dylan H. de Vries, Maria C. Jenmalm, Eva Landberg, Eva Sverremark-Ekström, Magalí Martí, Thomas Abrahamsson

**Affiliations:** 1Department of Biomedical and Clinical Sciences, Linköping University, SE-581 83 Linköping, Sweden; j.e.spreckels@umcg.nl (J.E.S.); erik.wejryd@liu.se (E.W.); maria.jenmalm@liu.se (M.C.J.); Eva.Landberg@regionostergotland.se (E.L.); magali.marti.genero@liu.se (M.M.); 2Department of Genetics, University Medical Centre Groningen, 9713 AV Groningen, The Netherlands; d.h.de.vries@umcg.nl; 3Department of Neonatology, Astrid Lindgren Children’s Hospital, Karolinska University Hospital and Institute, SE-171 76 Stockholm, Sweden; giovanna.marchini@sll.se (G.M.); baldvin.jonsson@ki.se (B.J.); 4Department of Clinical Chemistry, Linköping University, SE-581 83 Linköping, Sweden; 5Department of Molecular Biosciences, The Wenner-Gren Institute, Stockholm University, SE-106 91 Stockholm, Sweden; eva.sverremark@su.se; 6Department of Paediatrics, Linköping University, SE-581 83 Linköping, Sweden

**Keywords:** antibiotic, extremely low birth weight, feeding intolerance, human milk oligosaccharide, *Lactobacillus reuteri*, premature, probiotic, randomised controlled trial

## Abstract

*Lactobacillus reuteri* DSM 17938 supplementation reduces morbidities in very low birth weight infants (<1500 g), while the effect on extremely low birth weight infants (ELBW, <1000 g) is still questioned. In a randomised placebo-controlled trial (ClinicalTrials.gov ID NCT01603368), head growth, but not feeding tolerance or morbidities, improved in *L. reuteri*-supplemented preterm ELBW infants. Here, we investigate colonisation with the probiotic strain in preterm ELBW infants who received *L. reuteri* DSM 17938 or a placebo from birth to postmenstrual week (PMW) 36. Quantitative PCR was used on 582 faecal DNA samples collected from 132 ELBW infants at one, two, three, and four weeks, at PMW 36, and at two years of age. Human milk oligosaccharides were measured in 31 milk samples at two weeks postpartum. At least 86% of the ELBW infants in the *L. reuteri* group were colonised with the probiotic strain during the neonatal period, despite low gestational age, high antibiotic pressure, and independent of infant feeding mode. Higher concentrations of lacto-N-tetraose, sialyl-lacto-N-neotetraose c, and 6′-sialyllactose in mother’s milk weakly correlated with lower *L. reuteri* abundance. Within the *L. reuteri* group, higher *L. reuteri* abundance weakly correlated with a shorter time to reach full enteral feeding. Female sex and *L. reuteri* colonisation improved head growth from birth to four weeks of age. In conclusion, *L. reuteri* DSM 17938 supplementation leads to successful colonisation in ELBW infants.

## 1. Introduction

Mortality and morbidity of extremely preterm (birth before gestational week (GW) 28) extremely low birth weight (ELBW, <1000 g) infants born in Sweden has improved during the last decade [1] Still, 23% of the infants die during the first year of life, and 50% of the surviving infants suffer from major morbidities [1]. Meta-analyses suggest that probiotics reduce the incidence of necrotising enterocolitis (NEC), death, and NEC-related death in very low birth weight (VLBW, <1500 g) infants [2,3], but also that effects are strain-dependent [4]. One promising candidate strain is *Lactobacillus reuteri* DSM 17938. Three randomised controlled trials (RCTs) report beneficial effects of this strain for low birth weight infants. First, *L. reuteri* DSM 17938 was shown to improve feeding tolerance, reduce sepsis, and shorten the duration of hospitalisation of VLBW and ELBW infants born before GW 32 [5]. Second, supplementation of this *L. reuteri* strain shortened the time to full enteral feeding and the duration of hospitalisation of infants with a birth weight between 1500 and 2000 g [6]. Third, it also reduced feeding intolerance and tended to reduce NEC and to lower mortality in infants weighing between 1000 and 1800 g at birth [7]. In addition, a meta-analysis showed a reduction of NEC and a shorter time to full enteral feeding in preterm infants (born before GW 37) supplemented with *L. reuteri* DSM 17938 [4]. Importantly, this meta-analysis also pointed out that studies in the ELBW infant population are scarce. Only one of the three mentioned RCTs investigated the benefits of *L. reuteri* DSM 17938 specifically in ELBW infants [5]. The efficacy of probiotics such as *L. reuteri* DSM 17938 in the subgroup of ELBW infants thus remains questioned [4].

In order to investigate the effects of *L. reuteri* DSM 17938 in the most vulnerable preterm infants, our research group conducted a double-blind RCT using *L. reuteri* DSM 17938 only in extremely preterm ELBW infants. We did not find any effect on feeding intolerance, NEC, and sepsis, but head growth improved in the *L. reuteri*-supplemented group [8].

To date, it remains unclear why probiotics seem less effective in ELBW infants. Other factors than gut microbiota, for example, a reduced mucosal barrier function, hypoxic episodes, and impaired microvascular circulation, might play a larger role in the development of complications in ELBW as compared to VLBW infants [9]. High antibiotic pressure and immaturity of ELBW infant guts could also impair colonisation with probiotics and thus withhold them from exerting benefits [10,11]. The mother strain of *L. reuteri* DSM 17938, *L. reuteri* ATCC 55730, can colonise stomach, duodenum, and ileum of healthy adults [12], and viable *L. reuteri* are present in infant faeces after supplementation [13]. In line with this, we recently reported for our RCT that extremely preterm ELBW infants of the *L. reuteri*-supplemented group have high *L. reuteri* prevalence and abundance in faeces during the neonatal supplementation period [14]. Additionally, cross-colonisation of placebo-supplemented infants could mask potential beneficial effects of probiotics in preterm infants [11]. However, our recent analysis shows that cross-colonisation of the placebo group in our RCT was minor [14].

The aim of this study was to investigate how antibiotic treatment, infant feeding regimen, human milk oligosaccharides (HMOs), and maternal and infant characteristics affect colonisation of extremely preterm ELBW infants with probiotic *L. reuteri* DSM 17938. In addition, *L. reuteri* colonisation was linked to clinical outcomes of ELBW infants. Our hypotheses were that lower gestational age, antibiotic treatment, and breast milk feeding would negatively affect *L. reuteri* colonisation, and that *L. reuteri* colonisation would improve ELBW infant head growth. Here, we show that extremely preterm ELBW infants were successfully transiently colonised with probiotic *L. reuteri* DSM 17938, independent of gestational age, most antibiotic treatment, and feeding mode. *L. reuteri* colonisation improved head growth from birth to four weeks of age.

## 2. Materials and Methods

### 2.1. Study Design and Sample Collection

The PROPEL trial was a prospective, randomised, double-blind, placebo-controlled, multi-centre study on the effects of *L. reuteri* on feeding tolerance, growth, morbidities and mortality in extremely preterm ELBW infants (ClinicalTrials.gov ID NCT01603368). Ethical approvals were given by the Ethics Committee for Human Research in Linköping, Sweden (Dnr 2012/28–31, 22 February 2012; Dnr 2012/433–32, 23 January 2013). Written informed consent was obtained from parents within three days after delivery. A detailed study design has been published [8]. Briefly, the trial included 134 infants born between GW 23 + 0 and 27 + 6 with a birth weight below 1000 g, who were randomised to receive daily oral administration of 1.25 × 10^8^
*L. reuteri* DSM 17938 (BioGaia AB, Stockholm, Sweden) or a placebo (BioGaia AB) from one to three days postpartum until postmenstrual week (PMW) 36 (Figure 1). *L. reuteri* DSM 17938 was generated by removal of two resistance gene-carrying plasmids from its mother strain *L. reuteri* ATCC 55730 [15], which was originally isolated from the breast milk of a Peruvian mother [16]. The probiotic and placebo (maltodextrin) products were provided in oil drops consisting of sunflower oil, medium chain triglyceride oil, and silicon dioxide. The placebo could not be differentiated from the probiotic product by smell, taste, or visual appearance. Study products were administered to ELBW infants via a nasogastric tube or via the infant’s mouth if the nasogastric tube had been removed. Administration of the study products was withheld during periods in which infants were fed nil orally. After administration in the gastric tube, the drops with the study product were flushed down by at least 0.3 mL breast milk. All infants were exclusively fed mother’s own or donor milk until they reached a weight of at least 2000 g.

A total of 582 faecal samples were collected from ELBW infants at one, two, three, and four weeks, at PMW 36, and at two years of age. A total of 31 breast milk samples were collected at two weeks postpartum from mothers, whose infants were exclusively fed their own mother’s milk. Faeces and milk samples were stored at −20 °C (short-term) and subsequently at −80 °C until analysis.

### 2.2. Clinical Outcomes

Clinical characteristics were reported daily until PMW 36. Feeding tolerance was defined as the time to full enteral feeding (≥150 mL/kg/day). NEC was staged according to Bell’s criteria [17]; all cases of NEC stage II or greater were recorded. Sepsis was diagnosed if an infant had a positive blood and/or cerebral spinal fluid culture, clinical deterioration, and a laboratory inflammatory response. The culture method used at the hospitals was sensitive for *Lactobacilli*. Infant deaths were recorded. Bronchopulmonary dysplasia (BPD) was diagnosed if an infant required oxygen supplementation at 36 completed gestational weeks. Retinopathy of prematurity (ROP), intraventricular haemorrhage (IVH), and periventricular leukomalacia (PVL) were defined using international classifications [18,19,20]. Severe morbidities were defined as death, NEC stage II–III, sepsis, BPD, ROP grade 3–5, IVH grade 3–4, and/or PVL. Infant weight, length, and head circumference were measured at birth, at two, and at four weeks of age. For each growth measurement, the standard deviation score (z-score) was calculated using Niklasson’s growth chart [21], and growth rates were calculated as the difference in z-score between the later measurements and birth.

### 2.3. Lactobacillus reuteri Cultures

*L. reuteri* were cultured anaerobically in 10 mL De Man, Rogosa, and Sharpe (MRS) broth at 36 °C for 24 h. To detect viable *L. reuteri* in infant faeces, faecal dilutions were streaked on MRS agar plates with 2 µg/mL ampicillin (Sigma-Aldrich, Stockholm, Sweden) and 50 µg/mL vancomycin (Sigma-Aldrich), and incubated at 36 °C for 24 to 72 h.

### 2.4. DNA Extraction

DNA was extracted from 0.1 ± 0.03 g wet faeces using the QIAamp PowerFecal DNA kit on a QIAcube (Qiagen, Hilden, Germany) as described previously [14].

DNA was extracted from cultured *L. reuteri* with the EZ1 DNA Tissue kit (Qiagen). Broth cultures were centrifuged at 3000 rpm for 10 min at 4 °C, and pellets were resuspended in 500 µL buffer G2, while up to 10 colonies were picked from agar plates and pooled in 500 µL G2 buffer. Samples were transferred to glass bead tubes and shaken with a TissueLyser II for 1 min at 30 Hz. Two hundred µL lysate were subjected to automatised DNA extraction using the protocol for purification of DNA from bacterial culture samples on an EZ1 Advanced XL robot (Qiagen).

DNA concentration was measured with Qubit dsDNA HS Assay kits (Thermo Fisher Scientific, Waltham, MA, USA) according to the manufacturer’s instructions, and DNA was stored at −20 °C.

### 2.5. Quantitative PCR

The *L. reuteri* DSM 17938 strain-specific single-copy gene *Lactobacillus reuteri* unknown extracellular protein *lr1694* (GenBank accession number: DQ074924.1) was amplified using quantitative PCR (qPCR) (forward primer: 5’-TTAAGGATGCAAACCCGAAC-3’, reverse primer: 5’-CCTTGTCACCTGGAACCACT-3’; [22]). Each 20 µL reaction consisted of 2 µL of 10-fold diluted DNA, 1X SsoFast^TM^ EvaGreen^®^ Supermix (Bio-Rad, Hercules, CA, USA) and 300 nM of each primer. qPCR assays were performed in CFX96^TM^ Real-Time PCR Detection Systems (Bio-Rad) with the following program: 2 min at 98 °C, 40 cycles of 5 s at 98 °C, and 5 s at 63 °C. Melting curve analyses were performed after completed qPCR by increasing the temperature from 65 °C to 95 °C in 0.5 °C increments every 5 s. Standard curves with 2.5 × 10^1^ to 5 × 10^4^
*lr1694* gene copies per 1 µL corresponding to quantification limits of 2.3 × 10^4^ to 4.5 × 10^7^ gene copies per 1 µL were prepared by serially diluting DNA from cultured *L. reuteri*. Standard curve values (mean (95% confidence interval)) were similar in all runs (*n* = 24): slope = −3.51 (−3.55–−3.47), y-intercept = 42.3 (42.0–42.6), efficiency = 93% (91%–95%), and R^2^ = 0.996 (0.995–0.997). All samples, standards, and no template controls were run in duplicates. The qPCR was repeated for samples, for which the Cq values between duplicates differed more than 0.3. Non- or 20-fold-diluted samples were used if Cq values fell outside the standard curve range. Non-diluted samples with Cq values above 35 were considered negative for *L. reuteri*. Neither unspecific amplification nor PCR inhibition were observed. Results were normalised to faeces input used for DNA extraction and expressed as *L. reuteri* bacteria per 1 g wet faeces.

### 2.6. Human Milk Oligosaccharide Analysis

The concentrations of 15 major HMOs (Appendix A) were measured in milk after purification, using high-performance anion-exchange chromatography with pulsed amperometric detection, as described previously [23]. The Lewis and secretor status was based on the HMO profile (see Appendix A).

### 2.7. Statistical Analyses

The scientist performing the microbial analyses was blinded until the start of the statistical analysis. Fisher’s exact test was used to find differences in the distribution of study subjects to different categorical variables. Student’s *t*-test and Mann–Whitney *U* test were employed to detect differences between normally and non-normally distributed continuous data sets, respectively. The Kruskal–Wallis test with Dunn’s post hoc test was used to compare non-normally distributed *L. reuteri* levels at different time points. The method from Benjamini and Hochberg was used for multiple testing correction. Univariate and multivariate linear and logistic regression models were used to investigate the effects of covariates on linear and dichotomous outcomes, respectively. Spearman’s rank correlation was used for examining correlation between continuous variables. Statistical analyses were performed in R version 3.5.1 using the *FSA*, *Rmisc*, *PMCMRplus*, and *tidyr* packages, while graphs were created with the *ggplot2*, *ggpubr*, and *gridExtra* packages.

## 3. Results

### 3.1. Study Participants and Samples

The original RCT included 134 extremely preterm ELBW infants, of which 68 infants received probiotic *L. reuteri* and 66 infants received a placebo (Figure 1). For this study, 582 faecal samples collected at six time points from 132 of the 134 infants were analysed for the probiotic *L. reuteri* using qPCR. Potential confounders, i.e., clinical characteristics that significantly differed between the *L. reuteri*-supplemented and the placebo group, are shown in Appendix A for each time point, at which faecal samples were collected. Female sex was more common at one week, three weeks, and PMW 36, more infants were born by caesarean section at two weeks, three weeks, and PMW 36, more infants were from a multiple pregnancy at PMW 36, and more infants were born to mothers with chorioamnionitis at PMW 36 and two years of age in the *L. reuteri*-supplemented group compared to the placebo group.

### 3.2. Lactobacillus reuteri in Infant Faeces

During the neonatal period, *L. reuteri* prevalence and abundance were significantly higher in faeces of the *L. reuteri*-supplemented group than in faeces of the placebo group (Figure 2). All *L. reuteri*-supplemented infants had a faecal sample positive for the probiotic *L. reuteri* strain at least once, with the exception of one infant, which passed away before PMW 36 and for which faeces was only collected at one week of age. At two years of age, all but one faecal sample were negative for *L. reuteri*. Stratification of infants by sex, mode of birth, whether an infant was from a multiple pregnancy, and whether an infant was born to a mother with or without chorioamnionitis did not affect the results. Median *L. reuteri* abundance in faeces of the *L. reuteri*-supplemented group ranged from 1.90 × 10^7^ to 6.76 × 10^7^ bacteria per 1 g wet faeces in the first four weeks of life, while it was significantly lower at PMW 36 (median 7.09 × 10^6^) and two years of age (median 0) (Figure 2b).

To verify that *L. reuteri* bacteria in infant faeces were viable, we cultured *Lactobacilli* from 16 faecal samples collected from *L. reuteri*-supplemented infants and identified the probiotic strain with qPCR. Fifteen of 16 cultured samples were positive for the probiotic *L. reuteri* strain, even though two of the 16 samples were considered negative in the qPCR without the intermediate culturing step. The one sample that was negative for *L. reuteri* in the cultures was also negative in the qPCR-only assay.

### 3.3. Maternal and Infant Characteristics and Lactobacillus reuteri Colonisation

Clinical characteristics potentially affecting *L. reuteri* colonisation of the *L. reuteri*-supplemented group at one week of age and at PMW 36 are displayed in Table 1.

The number of hours with the probiotic product before faecal sample collection at one week of age was significantly higher in *L. reuteri*-colonised compared to non-colonised infants (Table 1a). Similarly, *L. reuteri* abundance at one week of age was positively correlated with the number of hours with the probiotic product (Spearman’s rho = 0.55, Appendix A). All *L. reuteri*-supplemented infants in Linköping, but only 78% of the *L. reuteri*-supplemented infants included in Stockholm, were colonised at one week of age (Table 1a). When infants were stratified by their inclusion site, both the effect on *L. reuteri* prevalence and the correlation with *L. reuteri* abundance were only significant in infants included in Stockholm, but not in infants included in Linköping (Appendix A).

*L. reuteri*-supplemented infants born to mothers experiencing preterm premature rupture of membranes (PPROM) had significantly higher levels of *L. reuteri* in faeces at one week of age (Appendix A). Of note, *L. reuteri* abundance in faeces at two weeks of age was not affected by PPROM, while *L. reuteri* abundance in faeces at three weeks of age was lower if a mother had PPROM (*p* = 0.04). None of the other characteristics in Table 1a were associated with *L. reuteri* prevalence or abundance at one, two, three, and four weeks of age (Appendix A, Table 1a and Appendix A).

The number of days without the probiotic product before PMW 36 was significantly higher in infants who were not colonised compared to infants who were colonised with *L. reuteri* at PMW 36 (Table 1b). None of the characteristics shown in Table 1b affected *L. reuteri* abundance at PMW 36 (Appendix A).

### 3.4. Antibiotic Treatment and Lactobacillus reuteri Colonisation

We examined whether antibiotics affected colonisation with *L. reuteri*. All infants in the *L. reuteri*-supplemented group received one or multiple antibiotics for at least one day during the first week of life (mostly aminoglycosides (100%) and benzylpenicillins (98%)), and 72% were still treated with antibiotics in week four (mostly vancomycin (43%), aminoglycosides (38%), and carbapenems (23%)) (Figure 3). Antibiotic treatment did not affect *L. reuteri* prevalence in faeces of the *L. reuteri*-supplemented group at one to four weeks of age (Appendix A). There was no significant difference in *L. reuteri* abundance in faeces of infants who were or were not treated with antibiotics for at least one day during the week preceding the faecal sampling (Figure 4, Appendix A), with one exception. Significantly lower *L. reuteri* abundance in faeces at three and four weeks of age was detected in carbapenem-treated compared to non-treated infants (Figure 4, Appendix A). The total number of days with antibiotics from birth to PMW 36 affected neither *L. reuteri* prevalence (Table 1b) nor abundance (Appendix A) at PMW 36.

To confirm viability of *L. reuteri* bacteria despite antibiotic treatment, 10 of the 16 abovementioned cultured faecal samples were collected from *L. reuteri*-supplemented infants who had been treated with antibiotics for seven days preceding the faecal sampling. The supplemented *L. reuteri* strain was cultured from all 10 samples.

### 3.5. Infant Feeding and Lactobacillus reuteri Colonisation

Furthermore, we studied whether infant feeding influenced *L. reuteri* colonisation. All infants were exclusively fed mother’s own or donor milk until they weighed at least 2000 g, thus investigating the effect of feeding mode on early *L. reuteri* colonisation was not possible. There was no significant difference in *L. reuteri* prevalence or abundance in faeces of *L. reuteri*-supplemented infants at PMW 36 when comparing formula-fed, partially breast milk-fed, and exclusively breast milk-fed infants (Figure 5).

### 3.6. Human Milk Oligosaccharides and Lactobacillus reuteri Colonisation

To investigate how Lewis and secretor status, and specific HMOs were associated to *L. reuteri* colonisation, we collected 31 milk samples at two weeks postpartum from the mothers of 36 exclusively mother’s own milk-fed infants from the *L. reuteri*-supplemented group. The concentrations of 15 HMOs at two weeks were used to determine a mother’s Lewis and secretor status (Appendix A), which was then associated to *L. reuteri* colonisation at three weeks of age. Neither a mother’s Lewis status nor her secretor status affected *L. reuteri* prevalence or abundance in faeces of the *L. reuteri*-supplemented group (Appendix A). Interestingly, two of the three infants with faeces negative for *L. reuteri* received milk from a non-secretor mother, while 9 of the 36 exclusively mother’s own milk-fed infants had a non-secretor mother.

Additionally, the concentrations of each of the 15 HMOs at two weeks of age were linked to *L. reuteri* colonisation data from three weeks of age. The 33 *L. reuteri*-supplemented infants with a *L. reuteri*-positive faecal sample at three weeks of age received mother’s milk with lower concentrations of lacto-N-tetraose (LNT) and 6′-sialyllactose (6′ SL) than the three non-colonised *L. reuteri*-supplemented infants (Appendix A). Higher concentrations of LNT, sialyl-lacto-N-neotetraose c (LSTc), and 6′ SL further weakly correlated with lower *L. reuteri* abundance in faeces at three weeks of age (Figure 6). There were no significant correlations for any other investigated HMOs (Figure 6).

### 3.7. Lactobacillus reuteri Colonisation and Clinical Outcomes

To investigate effects of *L. reuteri* colonisation on clinical outcomes, *L. reuteri*-colonised *L. reuteri*-supplemented infants were compared with non-colonised placebo infants (Figure 7a). *L. reuteri*-supplemented infants who were colonised with *L. reuteri* at three weeks of age had better head growth from birth until four weeks of age than non-colonised placebo-supplemented infants (median (95% confidence interval): −1.11 SD (−0.86–−1.35 SD) vs. −1.78 SD (−1.50–−2.06 SD), Figure 7e). Weight and length growth rates, the time to reach full enteral feeding, and incidences of severe morbidities were similar in the two groups (Appendix Ab–d,f–k). In multivariate models including the potential confounding factor sex and *L. reuteri* colonisation, female sex was significantly associated with improved length growth until four weeks of age (*p* = 0.007), and with head growth until two (*p* = 0.045), and four weeks of age (*p* = 0.013), while *L. reuteri* colonisation only affected head growth until four weeks of age (*p* = 0.003) (Appendix A). Differences in caesarean section frequency between groups did not affect the results.

Within the group of *L. reuteri*-colonised and non-colonised *L. reuteri*-supplemented infants, we further investigated whether *L. reuteri* prevalence and abundance at one week of age affected clinical outcomes. Significantly more *L. reuteri*-supplemented infants, who were colonised with *L. reuteri* at one week of age, had a time to reach full enteral feeding below or equal to the group’s median of 14 days (Figure 8a). Additionally, higher *L. reuteri* abundance in faeces at one week of age was significantly but weakly correlated with a shorter time to reach full enteral feeding (Figure 8b). *L. reuteri* abundance in faeces at one week of age was also significantly higher in infants who later developed culture-proven sepsis (Figure 8c). Of note, *L. reuteri* abundance in faeces at two, three, and four weeks of age was not associated with a higher risk of culture-proven sepsis. Other clinical outcomes were affected neither by *L. reuteri* prevalence nor *L. reuteri* abundance in faeces at one week of age.

## 4. Discussion

This RCT showed that extremely preterm ELBW infants can be successfully colonised with probiotic *L. reuteri* DSM 17938, and that *L. reuteri* colonisation occurred despite low gestational age, high antibiotic pressure, and independent of an infant’s feeding type.

ELBW infants received supplementation every day from one to three days after birth until PMW 36. During this supplementation period, at least 86% of the extremely preterm ELBW infants in the *L. reuteri*-supplemented group had a faecal sample positive for the probiotic. Since gastric and intestinal colonisation correlates with shedding of living *L. reuteri* in faeces [12], and *L. reuteri* could be cultured from a subset of infant faecal samples, we believe that detection of *L. reuteri* in the faecal samples indicated successful colonisation of the intestines of extremely preterm ELBW infants with the probiotic bacteria during the first weeks of life.

Antibiotic treatment was very common during the first four weeks of life in our study population. Previous reports showed that the probiotic *L. reuteri* DSM 17938 strain used in this study is resistant to several different antibiotics including aminoglycosides, beta-lactam antibiotics like ampicillin, and vancomycin [24]. In line with the resistance pattern of *L. reuteri*, antibiotics generally did not impede *L. reuteri* colonisation of extremely preterm ELBW neonates. Antibiotics were administered to extremely preterm ELBW infants intravenously, potentially leading to low concentrations of antibiotics in the infant guts, which then did not affect *L. reuteri* colonisation. However, treatment with carbapenems during the third and fourth week of life lowered *L. reuteri* abundance in extremely preterm ELBW infant faeces. Unfortunately, no previous reports on the resistance pattern of *L. reuteri* to carbapenems were available. We speculate that the probiotic *L. reuteri* could be sensitive to carbapenems or that carbapenem treatment alters the intestinal microenvironment in a way that hinders the growth of the supplemented *L. reuteri* strain. In a previous trial in full-term infants, the prevalence of *L. reuteri* in faeces after supplementation of the mother strain *L. reuteri* ATCC 55730 was not affected by antibiotic treatment during the first year of life, but no data specifically on carbapenems were shown [13]. A study by Rougé et al. (2009) [10] on ELBW infants and the PiPS study [11] on very preterm infants suggested that the colonisation with and effects of the supplemented strains *Bifidobacterium longum* BB536 and *Lactobacillus rhamnosus* GG or *Bifidobacterium breve* BBG-001, respectively, were hindered by antibiotic treatment. Their findings are in contrast to the general results from our study, which might be explained by different antibiotic resistance patterns of the probiotic strains. Unfortunately, data on different antibiotic classes were not presented in the two studies, so it remains unclear which antibiotic class(es) hindered probiotic growth in these previous studies. Even though *L. reuteri* DSM 17938 might be susceptible to carbapenems in our study, it is remarkable how well the supplemented *L. reuteri* bacteria tolerate the (broad-spectrum) antibiotics commonly administered to extremely preterm ELBW infants.

The abundance of *L. reuteri* in faeces of extremely preterm ELBW infants was high during the neonatal period, and decreased at PMW 36, despite the same dose being administered. *L. reuteri* was not detected in faeces of the *L. reuteri*-supplemented group at two years of age, when the infants no longer received the probiotic product, confirming that *Lactobacillus* strains merely colonise the infant gut transiently. Possibly, *L. reuteri* is outcompeted by other colonisers when the gut microbiota matures [14,25]. In contrast to the PiPS trial [11], low gestational age did not negatively affect colonisation in this study. *L. reuteri* colonisation at one week of age was dependent on the number of hours that an infant received the probiotic product between birth and faecal sampling at one week, and *L. reuteri* prevalence at PMW 36 was lower if infants had more days between birth and PMW 36 without the probiotic product. These findings highlight that *L. reuteri* colonisation depends on regular *L. reuteri* intake, which is in agreement with Rougé et al. [10], who described that suspension of enteral feeding and the associated suspension of probiotic administration possibly lead to a lack of colonisation of ELBW infants with *B. longum* BB536 and *L. rhamnosus* GG.

Even though breast milk feeding lowered *L. reuteri* abundance in faeces in a previous trial in full-term infants [13] who received *L. reuteri* ATCC 55730, the mother strain of the probiotic used in this study, we found no negative effect of any feeding mode on *L. reuteri* colonisation in extremely preterm ELBW infants. In the previous trial, however, *L. reuteri* was also given to mothers during the last four weeks of pregnancy, and *L. reuteri* was detected in 12% of the colostrum samples [13]. Infants might thus have received breast milk with antibodies against the probiotic strain, potentially hindering colonisation with *L. reuteri*. The lack of effect of feeding mode on *L. reuteri* colonisation in our RCT is also in contrast to the PiPS trial [11], in which breast milk reduced colonisation of very preterm infants with probiotic *B. breve* BBG-001. Interestingly, in bacterial cultures, *L. reuteri* DSM 17938 growth was enhanced by human breast milk compared to formula [26].

Although *L. reuteri* colonisation was not affected by feeding mode in our study, we were interested in whether variations in breast milk components such as HMOs affected *L. reuteri* colonisation. In a subset of 36 *L. reuteri*-supplemented infants, we detected a trend towards lower *L. reuteri* abundance in infants who received milk with higher levels of LNT, LSTc, and 6′ SL. Interestingly, two of the three infants who were not colonised with *L. reuteri* at three weeks of age received milk from non-secretor mothers, who cannot synthesise LNDH I and LNFP I, and thus have higher concentrations of their precursor LNT [27]. Previous in vitro studies showed that *L. reuteri* cannot metabolise most HMOs, including LNT and 6′ SL [28,29], hence it is not surprising that there is no growth-promoting effect of those HMOs on probiotic *L. reuteri*. Other bacteria, for example, certain *Bifidobacteria* strains, are able to metabolise HMOs [28] and might have a competitive advantage over non-metabolisers like the supplemented *L. reuteri*. Our results from associating HMO concentrations with *L. reuteri* colonisation give a first insight based on a small sample only. More research into studying the interactions between breast milk components like HMOs and different microbes and metabolites produced from microbial fermentation of milk sugars is needed to further understand this complex ecosystem and to ultimately provide good growth conditions for probiotic bacteria.

Besides poor colonisation of the probiotic group, cross-colonisation of the placebo group with the probiotic bacteria has been suggested as an explanation for a lack of effects of probiotics in preterm infants [11]. In our study, probiotic *L. reuteri* were detected in a maximum of 10% of the faecal samples of the placebo group. Moreover, comparison of clinical outcomes between *L. reuteri*-colonised *L. reuteri*-supplemented and non-colonised placebo-supplemented infants confirms the lack of effect described in the original trial that compared *L. reuteri* and placebo supplementation groups [8]. In line with our recent report on low *L. reuteri* prevalence and abundance in placebo-supplemented infants [14], this suggests that it is unlikely that cross-colonisation of placebo-supplemented infants caused the lack of effects of *L. reuteri* supplementation in our trial.

Interestingly, within the *L. reuteri*-supplemented subgroup, the primary outcome of the trial, the total time to reach full enteral feeding, was shorter in infants who had a higher *L. reuteri* abundance at one week of age. This suggests that *L. reuteri* abundance and thus the dose of the *L. reuteri* strain in the probiotic product are important, and that there might have been a stronger clinical effect if a higher dose had been administered. However, when investigating the effects of *L. reuteri* abundance on clinical outcomes in the *L. reuteri*-supplemented group, we also detected a potential increase in culture-proven sepsis in infants who had a higher *L. reuteri* abundance in faeces at one week of age. This effect was seen neither in the intention-to-treat analysis in the original publication on the clinical outcomes of these extremely preterm ELBW infants [8] nor when comparing non-colonised placebo and *L. reuteri*-colonised *L. reuteri*-supplemented ELBW infants in the current study. Importantly, there were no cases of sepsis with the probiotic *L. reuteri*. Our RCT was not powered to detect differences in sepsis incidence, and the analysis of sepsis occurrence was based on a subset of only 53 ELBW infants. *L. reuteri* supplementation has further been described to reduce sepsis in a previous RCT in 196 ELBW infants [5] using a similar dose as in our trial (1 × 10^8^ vs. 1.25 × 10^8^ bacteria per day in our RCT). Although our results should be interpreted with caution, a potentially shorter time to reach full enteral feeding and a potentially increased sepsis incidence with higher *L. reuteri* abundance indicate that changing treatment doses in extremely preterm ELBW infants could have both positive and negative effects on clinical outcomes. In a previous trial, infants with a birth weight below 750 g were more likely to suffer from sepsis than infants with a birth weight between 750 and 1500 g [30], and it is thus important to remain cautious with the administration of probiotics to the most vulnerable extremely preterm ELBW infants.

In line with the first paper about this trial [8], we describe here an improved head growth in female and *L. reuteri*-colonised extremely preterm ELBW infants. Since *L. reuteri* colonisation and higher *L. reuteri* abundance shortened the time to full enteral feeding within the *L. reuteri*-supplemented infant group, it is tempting to speculate that *L. reuteri* improved the infants’ nutrition and thereby positively affected their growth. The gut microbiota can influence the peripheral and central nervous system and host metabolism via production of neurotransmitters and short chain fatty acids [31,32]. A study in undernourished Bangladeshi infants at 12 to 18 months of age described changes in concentrations of blood plasma proteins linked to neurodevelopment due to manipulation of the gut microbiota [33]. Although the head growth of the infants in our trial was still inferior to the growth in utero, an increased head circumference growth rate might be associated with improved neurodevelopment of extremely preterm infants [34]. Intake of *L. reuteri* during the neonatal period might thus prevent or ameliorate neurological disabilities frequently present in premature infants [35]. Indeed, supplementation of *L. reuteri* ATCC 55730, the mother strain of *L. reuteri* DSM 17938, positively affected neurological outcomes of preterm infants born before GW 37 with a birth weight below 2500 g [36]. However, another study reported no effect of *L. reuteri* DSM 17938 on neuromotor, neurosensory, and cognitive outcomes of VLBW infants at 18 to 24 months corrected age [37]. To ascertain whether the improved head growth detected in this study is of clinical importance, neurological outcome assessment of study participants at an older age is planned.

Importantly, this study was designed and powered to detect significant effects of *L. reuteri* supplementation on time to reach full enteral feeding in extremely preterm ELBW infants, while it was not powered to detect significant effects on secondary outcomes like NEC, sepsis, and mortality. Larger studies on extremely preterm ELBW infants are needed to verify the findings for these outcomes. As our RCT intentionally only included ELBW infants, this limited the possibility to generalise findings to other infant groups. The qPCR-based investigation of probiotic colonisation strengthened the results from our RCT, although it is important to note that detection of DNA from the probiotic *L. reuteri* strain in faeces does not equal successful intestinal colonisation. However, invasive faecal sampling from ELBW infant intestines was not possible since it would have been considered unethical. To verify that *L. reuteri* gene copies detected by qPCR in infant faeces did not solely derive from DNA of dead bacteria but rather represented viable bacteria, we confirmed our results with conventional cultivation for a subset of samples. By being able to culture probiotic *L. reuteri* bacteria from faeces of infants who had been treated with antibiotics, we could further verify that probiotic *L. reuteri* were indeed able to survive in the ELBW infants’ intestines despite exposure to antibiotics. Lastly, infants were included in our RCT in two different hospitals, which allowed to detect differences between hospitals, and helped to reduce the risk of presenting hospital-specific findings as general findings.

## 5. Conclusions

In conclusion, *L. reuteri* supplementation of extremely preterm ELBW infants resulted in a colonisation rate of at least 86% during the neonatal period, despite low gestational age, extensive antibiotic treatment, and independent of infant feeding mode. *L. reuteri* colonisation was transient, and all infants of the *L. reuteri* group were negative for the probiotic strain at two years of age. In the *L. reuteri*-supplemented group, a high *L. reuteri* abundance shortened the time to reach full enteral feeding, and higher concentrations of LNT, LSTc, and 6′ SL in mother’s milk weakly correlated with lower *L. reuteri* abundance in faeces. *L. reuteri* colonisation at three weeks of age improved head growth from birth to four weeks of age.

## Figures and Tables

**Figure 1 microorganisms-09-00915-f001:**
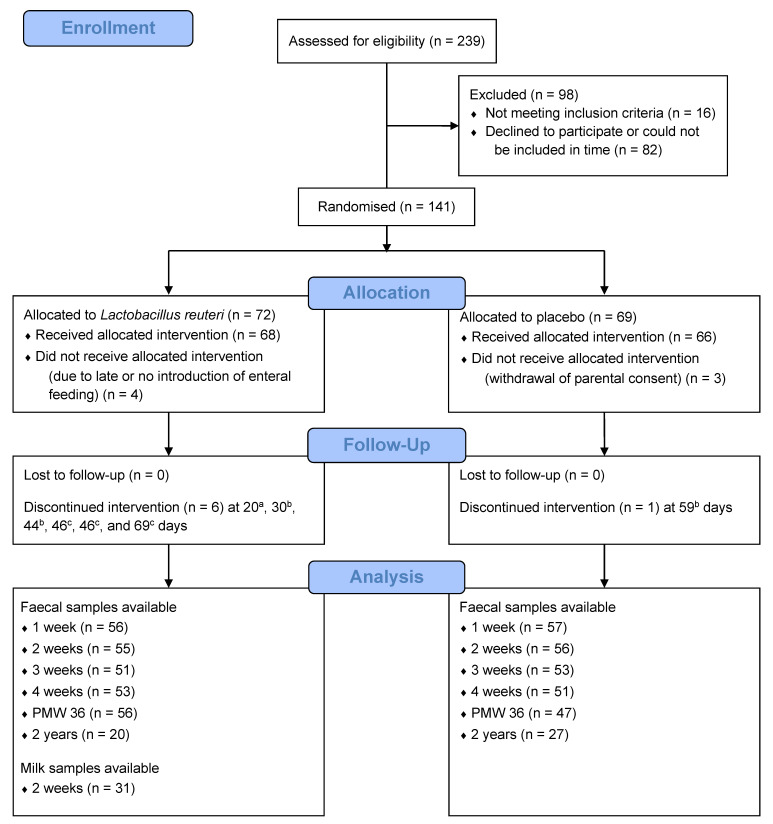
Flow chart of the PROPEL trial. PMW 36 = postmenstrual week 36. ^a^ Study product was discontinued by mistake after transfer to another hospital (*n* = 1). ^b^ Study product was not administered by mistake after temporarily being withheld during nil oral (*n* = 3). ^c^ Study product ran out temporarily at the study site (*n* = 3).

**Figure 2 microorganisms-09-00915-f002:**
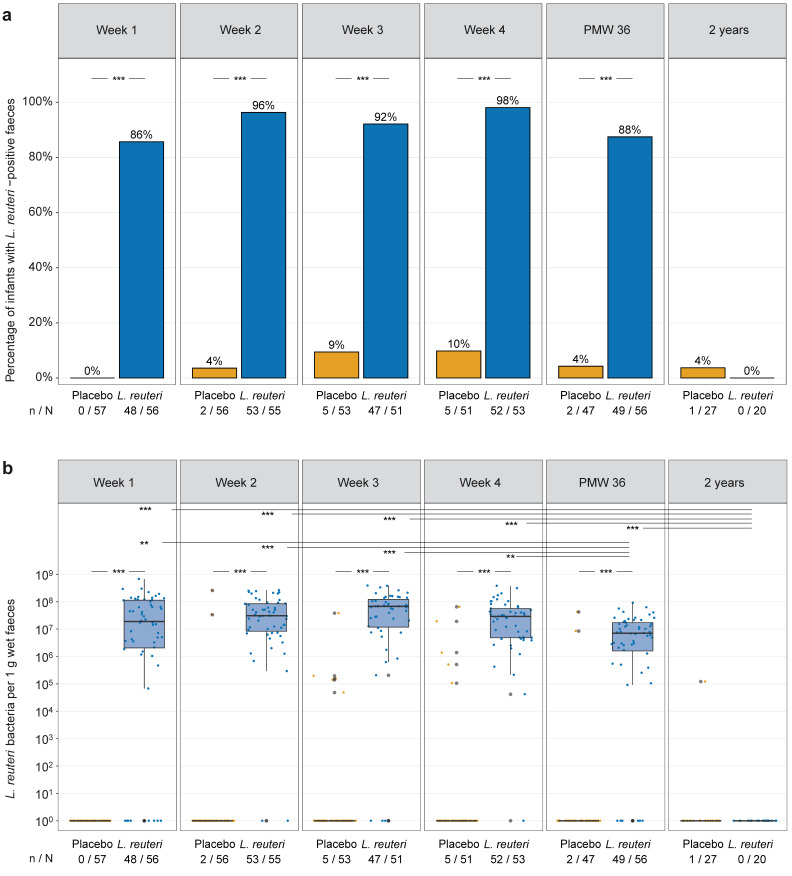
*L. reuteri* in faeces of the *L. reuteri*-supplemented and placebo group. *L. reuteri* prevalence (**a**) and abundance (**b**) in faeces of extremely preterm ELBW infants. Prevalence is expressed as the percentage of infants with a *L. reuteri*-positive faecal sample. Abundance is shown as the number of *L. reuteri* bacteria per 1 g wet faeces. Boxplots show median with 25% and 75% percentiles and 1.5× the interquartile range. Numbers (n / N) indicate the number of infants with a *L. reuteri*-positive faecal sample and the total number of infants per group. For graphical display, the number of *L. reuteri* bacteria per 1 g wet faeces in (**b**) was set to 10^0^ for infants with *L. reuteri*-negative faeces. Statistics: Fisher’s exact test with Benjamini–Hochberg correction (**a**), Mann–Whitney *U* test with Benjamini–Hochberg correction (comparisons between supplementation groups) and Kruskal–Wallis test with Dunn’s post hoc test with Benjamini–Hochberg correction (comparison of *L. reuteri* abundance in the *L. reuteri*-supplemented group over time) (**b**). ** adjusted *p* < 0.01, *** adjusted *p* < 0.001. ELBW = extremely low birth weight, PMW = postmenstrual week.

**Figure 3 microorganisms-09-00915-f003:**
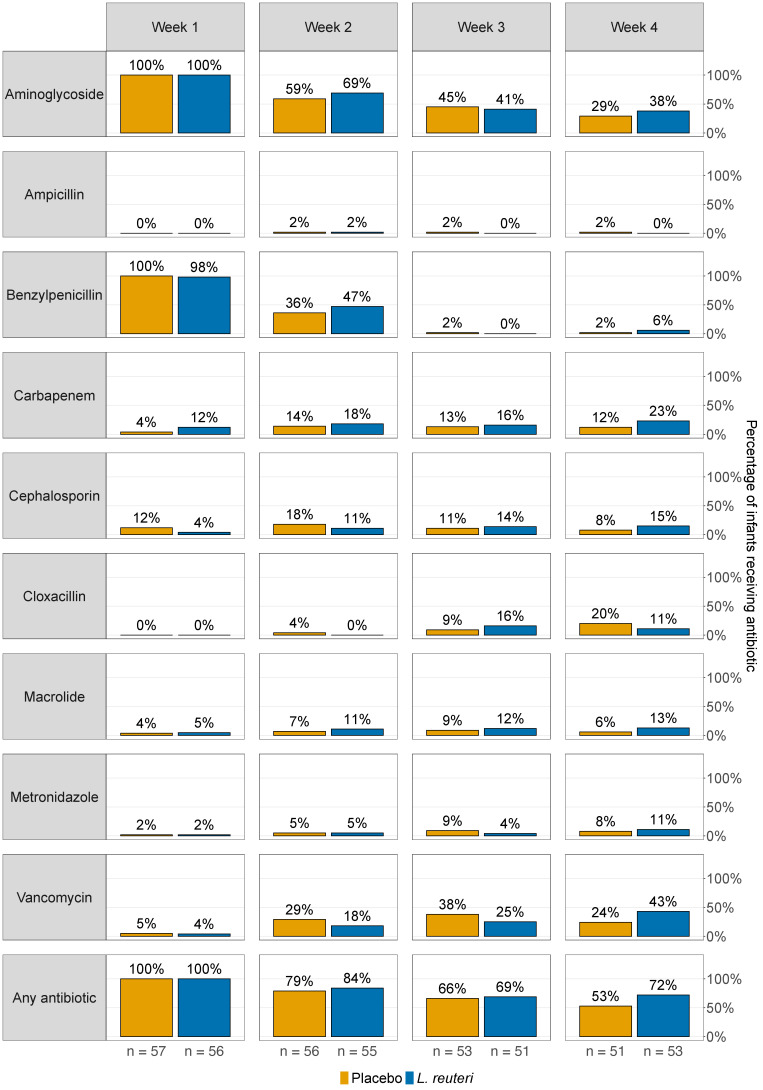
Antibiotic treatment in the *L. reuteri*-supplemented and placebo group. Bars show antibiotic treatment rates (in %) for placebo (yellow) and *L. reuteri*-supplemented (blue) extremely preterm ELBW infants, who were treated with the specified antibiotics for at least one day during the first, second, third, and fourth week of life. ELBW = extremely low birth weight.

**Figure 4 microorganisms-09-00915-f004:**
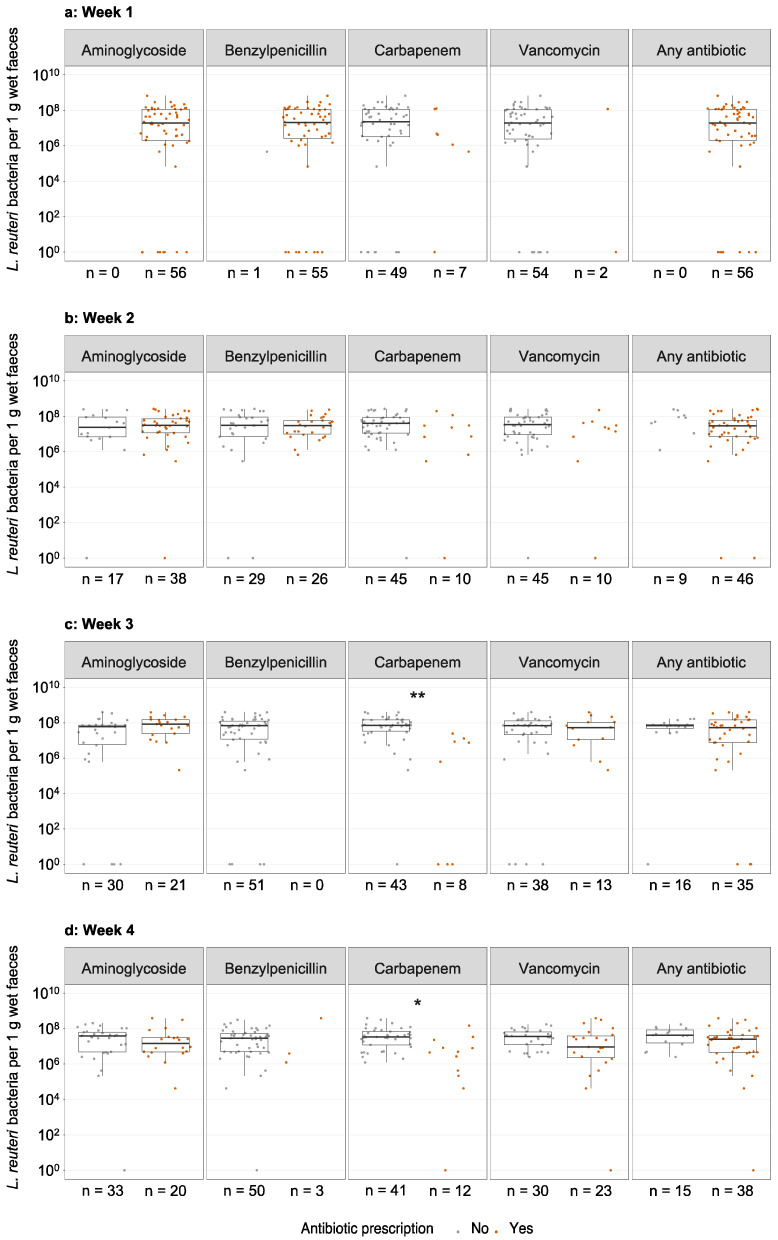
Antibiotic treatment and *L. reuteri* abundance in faeces of the *L. reuteri*-supplemented group. *L. reuteri* abundance in faeces of *L. reuteri*-supplemented extremely preterm ELBW infants at one (**a**), two (**b**), three (**c**), and four (**d**) weeks of age, who were not (grey dots) or were (red dots) treated with antibiotics for at least one day during the week preceding faecal sampling. Abundance is expressed as *L. reuteri* bacteria per 1 g wet faeces. Boxplots show median with 25% and 75% percentiles and 1.5× the interquartile range. For graphical display, the number of *L. reuteri* bacteria per 1 g wet faeces was set to 10^0^ for infants with *L. reuteri*-negative faeces. Numbers below the panels indicate the number of infants in the respective group. This figure only shows antibiotics, for which the sample size at any week was at least 10, while data on less common antibiotics are presented in Appendix A. Statistics: Mann–Whitney *U* test with Benjamini–Hochberg correction. * adjusted *p* < 0.05, ** adjusted *p* < 0.01. ELBW = extremely low birth weight.

**Figure 5 microorganisms-09-00915-f005:**
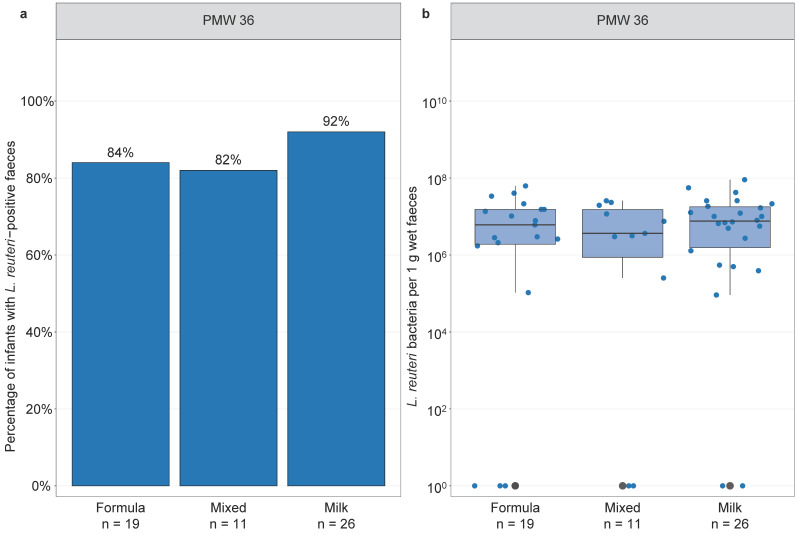
Infant feeding and *L. reuteri* colonisation in the *L. reuteri*-supplemented group. *L. reuteri* prevalence (**a**) and abundance (**b**) in faeces of *L. reuteri*-supplemented extremely preterm ELBW infants by feeding type. Prevalence is expressed as the percentage of infants with a *L. reuteri*-positive faecal sample and the total number of infants in each feeding category. Abundance is shown as the number of *L. reuteri* bacteria per 1 g wet faeces. Boxplots show median with 25% and 75% percentiles and 1.5× the interquartile range. For graphical display, the number of *L. reuteri* bacteria per 1 g wet faeces in (**b**) was set to 10^0^ for infants with *L. reuteri*-negative faeces. Infants in the partially breast milk-fed (‘mixed’) feeding group received 1–50% breast milk. Statistics: Fisher’s exact test (**a**), Kruskal–Wallis test (**b**). ELBW = extremely low birth weight, PMW = postmenstrual week.

**Figure 6 microorganisms-09-00915-f006:**
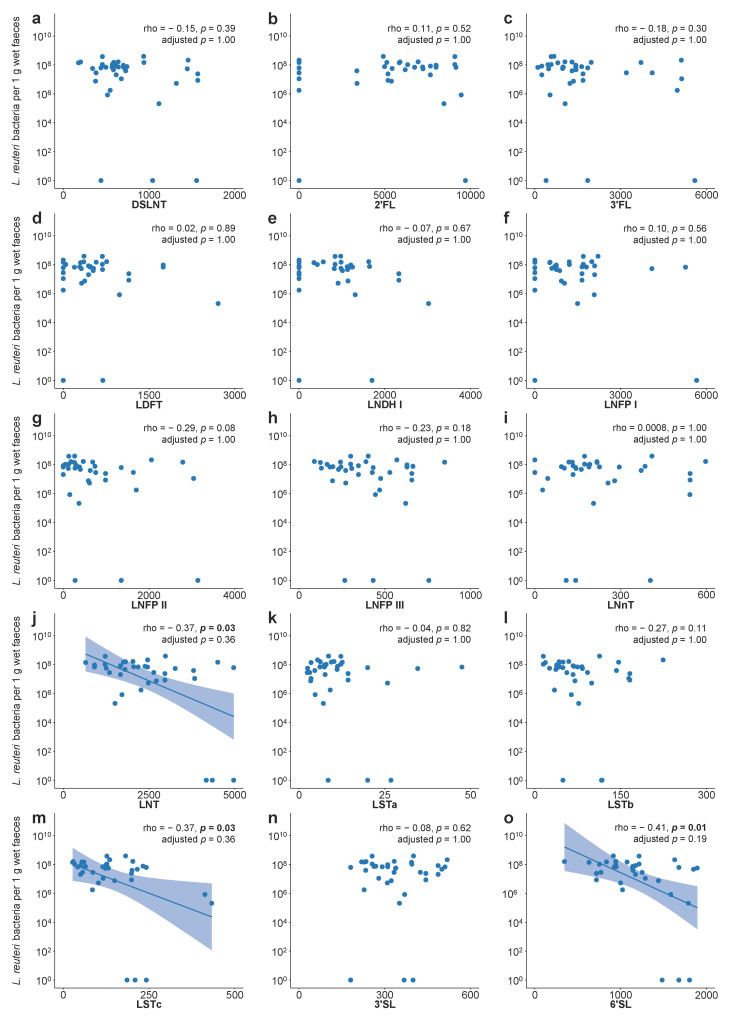
Human milk oligosaccharides and *L. reuteri* abundance in faeces of the *L. reuteri*-supplemented group. (**a**–**o**) Spearman correlation between human milk oligosaccharide concentrations (µmol/L) in milk collected at two weeks postpartum from exclusively breastfeeding mothers (*n* = 31) and *L. reuteri* abundance in faeces at three weeks of age (*n* = 36) from *L. reuteri*-supplemented infants. Abundance is shown as the number of *L. reuteri* bacteria per 1 g wet faeces. For infants with a faecal sample negative for *L. reuteri*, the number of *L. reuteri* bacteria per 1 g wet faeces was set to 10^0^. *p* values were adjusted with the method from Benjamini and Hochberg. DSLNT = disialyl-lacto-N-tetraose, 2′ FL = 2′-fucosyllactose, 3′ FL = 3′-fucosyllactose, LDFT = lacto-difucotetraose, LNDH I = lacto-N-difucohexaose I, LNFP I = lacto-N-fucopentaose I, LNFP II = lacto-N-fucopentaose II, LNFP III = lacto-N-fucopentaose III, LNnT = lacto-N-neotetraose, LNT = lacto-N-tetraose, LSTa = sialyl-lacto-N-tetraose a, LSTb = sialyl-lacto-N-tetraose b, LSTc = sialyl-lacto-N-neotetraose c, 3′ SL = 3′-sialyllactose, 6′ SL = 6′-sialyllactose.

**Figure 7 microorganisms-09-00915-f007:**
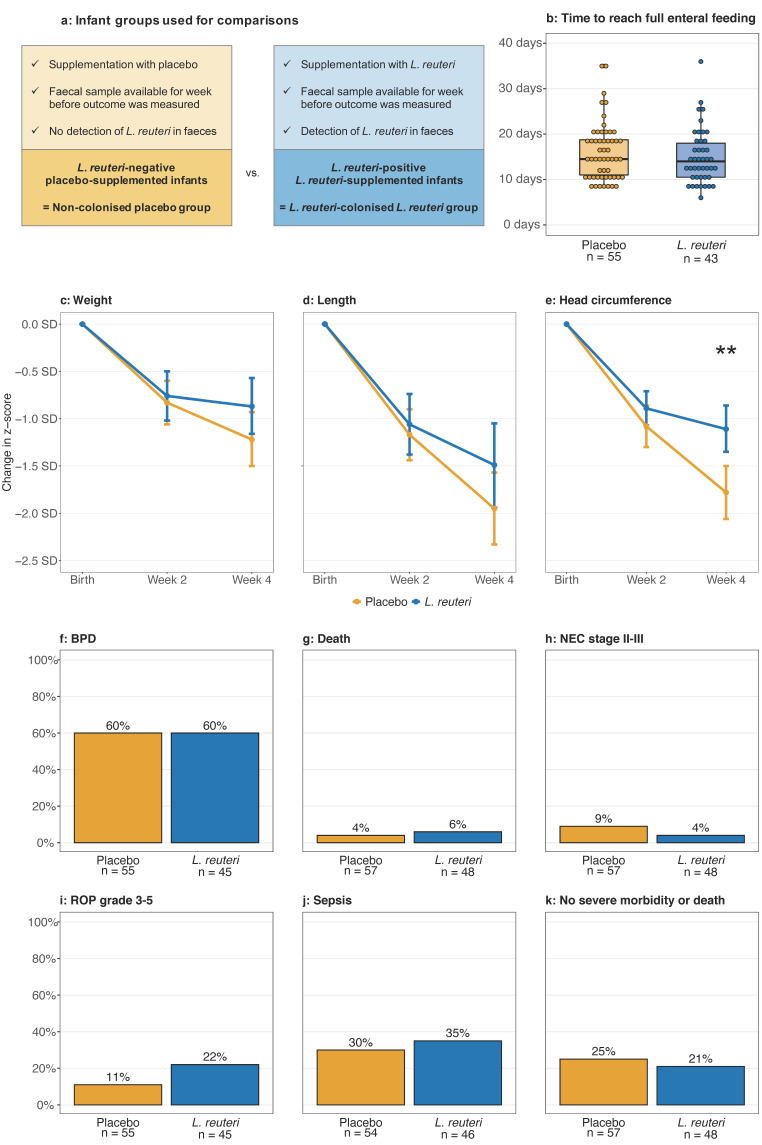
Growth and clinical outcomes in the *L. reuteri*-colonised *L. reuteri*-supplemented and non-colonised placebo group. (**a**) Criteria for selection of infants from the placebo and *L. reuteri*-supplemented groups for inclusion in the comparisons. (**b**) Boxplots (median with 25% and 75% percentiles and 1.5× the interquartile range) show the time to full enteral feeding of placebo and *L. reuteri*-supplemented infants with a faecal sample negative or positive for *L. reuteri* at one week of age, respectively. (**c**–**e**) Line graphs show the change in weight, length, and head circumference z-scores (mean with 95% confidence interval) from birth to two and four weeks of age in placebo and *L. reuteri*- supplemented infants with a faecal sample negative or positive for *L. reuteri* one week prior to the growth measurement, respectively. (**f**–**k**) Bars show prevalence of clinical outcomes of placebo and *L. reuteri*-supplemented infants with a faecal sample negative or positive for *L. reuteri* at one week of age, respectively. There were no significant differences in the prevalence of the clinical outcomes if infants were grouped based on *L. reuteri* colonisation at two, three, or four weeks of age. Infants were excluded if data on the specified outcome were missing or if outcomes occurred before or on the day of faecal sampling. One infant from the placebo group did not achieve full enteral feeding by PMW 36. This infant was included in the statistical analysis but is not shown in the plot in (**b**). Statistics: Mann–Whitney *U* test (**b**), Student’s *t*-test with Benjamini–Hochberg correction (**c**–**e**), Fisher’s exact test (**f**–**k**). ** adjusted *p* < 0.01. BPD = bronchopulmonary dysplasia, NEC = necrotising enterocolitis, PMW = postmenstrual week, ROP = retinopathy of prematurity.

**Figure 8 microorganisms-09-00915-f008:**
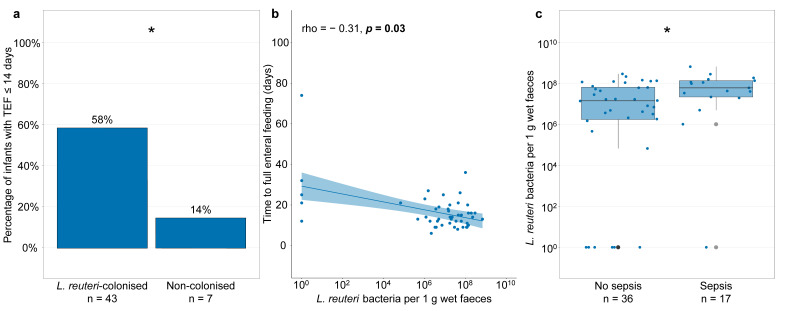
*L. reuteri* abundance in faeces of the *L. reuteri*-supplemented group at one week of age and time to full enteral feeding and sepsis. (**a**) Bars show percentage of infants with a time to full enteral feeding (TEF) below or equal to the group’s median of 14 days in infants of the *L. reuteri*-supplemented group, who had a faecal sample positive (*L. reuteri*-colonised) or negative (non-colonised) for *L. reuteri* at one week of age. (**b**) Spearman correlation between *L. reuteri* abundance at one week of age and TEF (expressed in days after birth) in *L. reuteri*-supplemented infants. (**c**) *L. reuteri* abundance in faeces of *L. reuteri*-supplemented infants at one week of age who later do or do not develop sepsis. Boxplots show median with 25% and 75% percentiles and 1.5× the interquartile range. Infants were excluded if information on TEF was missing (*n* = 3) or if infants reached full enteral feeding before faecal sampling at one week of age (*n* = 3) (**a**,**b**). For one infant who did not reach full enteral feeding by PMW 36, for graphical display, TEF was set to the day on which the infant reached PMW 36 (74 days) (**b**). Three infants were excluded because sepsis occurred before faecal sampling at one week of age (**c**). *L. reuteri* abundance is shown as the number of *L. reuteri* bacteria per 1 g wet faeces and, for graphical display, the number of *L. reuteri* bacteria per 1 g wet faeces was set to 10^0^ for infants with *L. reuteri*-negative faeces (**b**,**c**). Statistics: Fisher’s exact test (**a**), Mann–Whitney *U* test (**c**). * *p* < 0.05.

**Table 1 microorganisms-09-00915-t001:** Maternal and infant characteristics and *L. reuteri* prevalence in faeces of the *L. reuteri*-supplemented group. *L. reuteri*-supplemented infants were separated into colonised and non-colonised infants depending on whether their faeces were positive or negative for probiotic *L. reuteri* at one week of age (**a**) or at postmenstrual week 36 (**b**), respectively.

**a: Week 1**					
	***L. reuteri*-Colonised** **(*n* = 48)**		**Non-Colonised** **(*n* = 8)**		***p* Value**
Gestational age, weeks, mean (SD)	25.4	(1.3)	25.8	(1.1)	0.43 ^1^
Gestational weeks 23–25, n (%)	31	(65%)	4	(50%)	0.46 ^2^
Birth weight, g, mean (SD)	728	(130)	702	(142)	0.65 ^1^
Birth weight, z-score, mean (SD)	−1.2	(1.2)	−1.8	(1.1)	0.17 ^1^
Small for gestational age (weight < 2 SD), n (%)	13	(27%)	4	(50%)	0.23 ^2^
Apgar score at 5 min, median (IQR)	6.5	(4.0–8.0)	8	(4.0–8.2)	0.65 ^3^
Male/Female, n/n (%/%)	22/26	(46%/54%)	2/6	(25%/75%)	0.44 ^2^
Infant from multiple pregnancy, n (%)	16	(33%)	3	(38%)	1.00 ^2^
Caesarean section, n (%)	36	(75%)	6	(75%)	1.00 ^2^
Chorioamnionitis, n (%)	14	(29%)	2	(25%)	1.00 ^2^
Preeclampsia, n (%)	3	(6%)	2	(25%)	0.14 ^2^
Preterm premature rupture of membranes, n (%)	19	(40%)	2	(25%)	0.70 ^2^
Maternal smoking, n (%)	4	(8%)	0	(0%)	1.00 ^2^
Maternal antibiotics, n (%)	32	(67%)	3	(38%)	0.14 ^2^
Antenatal corticosteroids, n (%)	47	(98%)	8	(100%)	1.00 ^2^
Full course of antenatal corticosteroids, n (%)	31	(65%)	8	(100%)	0.09 ^2^
Inclusion site—Linköping/Stockholm, n/n (%/%)	19/29	(40%/60%)	0/8	(0%/100%)	0.04 ^2^
Start of supplementation within 72 h, n (%) ^a^	41	(85%)	7	(88%)	1.00 ^2^
Age when receiving first dose, h, mean (SD)	47	(20)	56	(21)	0.29 ^1^
Total time with probiotic product, h, mean (SD) ^b,c^	117	(39)	64	(30)	0.005 ^1^
**b: Postmenstrual Week 36**					
	***L. reuteri*** **-Colonised** **(*n* = 49)**		**Non-Colonised** **(*n* = 7)**		***p*** **Value**
Total time without probiotic product, days, median (IQR)	1	(0–3)	27	(13–39)	0.02 ^3^
Total time with acid inhibitors, days, median (IQR)	0	(0–5)	5	(0–10)	0.35 ^3^
Total time with antibiotics, days, median (IQR)	26	(20–37)	19	(17–52)	0.86 ^3^
Total time with insulin, days, median (IQR)	0	(0–1)	0	(0–1)	0.86 ^3^
Total time with continuous opioids, days, median (IQR)	5	(0–16)	3	(0–10)	0.79 ^3^
Total time with opioid antagonists, days, median (IQR)	0	(0–16)	0	(0–4)	0.36 ^3^
Total time with postnatal steroids, days, median (IQR)	0	(0–3)	0	(0–0)	0.48 ^3^
Total time with postnatal inhaled steroids, days, median (IQR) ^d^	24	(5–39)	20	(20–38)	0.76 ^3^

^a^ Data for one *L. reuteri*-colonised infant are missing. ^b^ Data for four *L. reuteri*-colonised infants are missing. ^c^ Data for two non-colonised infants are missing. ^d^ Data for one non-colonised infant are missing. Statistics: ^1^ Student’s *t*-test, ^2^ Fisher’s exact test, ^3^ Mann–Whitney *U* test. IQR = interquartile range, SD = standard deviation.

## Data Availability

The data presented in this study are available on request from the corresponding author. The data are not publicly available due to the General Data Protection Regulation (GDPR).

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
