# Peer review of "Lactobacillus reuteri Colonisation of Extremely Preterm Infants in a Randomised Placebo-Controlled Trial"

_microorganisms, 2021, doi:10.3390/microorganisms9050915_

Round 1

Reviewer 1 Report

L. reuteri supplementation was applied in preterm ELBW newborns and showed a colonisation rate of   86% . Yet,  a high L. reuteri abundance shortened the time to reach full enteral feeding, and higher concentrations of LNT, LST c, and 6’SL in mother’s milk weakly correlated with lower L. reuteri abundance in feces.  L. reuteri colonisation at three weeks of age improved head growth from birth to four weeks of age.

This is a well written paper with extended bibliography.

The research is well designed. The PROPEL trial which is a prospective, randomised, double-blind, placebo-controlled, multi-centre study on the effects of L. reuteri on feeding tolerance, growth, morbidities and 
mortality in extremely preterm ELBW infants (ClinicalTrials.gov ID NCT01603368) was used.

My suggestion is to ACCEPT it and publish in its present form.

Author Response

Comments from reviewer 1

  1. reuteri supplementation was applied in preterm ELBW newborns and showed a colonisation rate of   86% . Yet,  a high L. reuteri abundance shortened the time to reach full enteral feeding, and higher concentrations of LNT, LST c, and 6’SL in mother’s milk weakly correlated with lower L. reuteri abundance in feces.  L. reuteri colonisation at three weeks of age improved head growth from birth to four weeks of age.

This is a well written paper with extended bibliography.

The research is well designed. The PROPEL trial which is a prospective, randomised, double-blind, placebo-controlled, multi-centre study on the effects of L. reuteri on feeding tolerance, growth, morbidities and mortality in extremely preterm ELBW infants (ClinicalTrials.gov ID NCT01603368) was used.

My suggestion is to ACCEPT it and publish in its present form.

Response to reviewer 1

We want to thank the reviewer for the review, the kind words, and for suggesting the acceptance of our paper.

Reviewer 2 Report

The authors Spreckels et al. have investigated colonization of  Lactobacillus reuteri DSM 17938 in preterm ELBW infants from birth to postmenstrual week (PMW) 36 in a randomized manner and looked at body composition, health metrics. They not only found successful colonization of L. reuteri but also this colonization improved head growth from birth to four weeks of age. 

Comments:

Why did the authors specifically choose L. reuteri and ? There are multiple strains of probiotics in market, in testing phases and isolation. Each strain works in a individual way. The probiotic benefits may depend on it survival of gastric environment and reaching in enough numbers. 

The authors in their previous research (reference# 6) have shown oral supplementation with oil drops. Did the authors test if the oil have any effect on probiotic efficacy? is synergistic effect ? Where any stability studies conducted ? 

It would have been beneficial to compare with a standard or already available L. acidophilus or Bifido strains? 

What is the source of isolation of the probiotic strain?

Figure 2, colonization of L. reuteri, the authors showed the presence of L. reuteri from 2nd weeks placebo groups as well, but not first week, was there any correlation and influence of diet & feeding?

Looking at microbiome and showing the changes in the microbial communities specifically different Lactobacillus species in the gut w/ or w/o probiotic supplementation would be beneficial. 

Author Response

Comments from reviewer 2

The authors Spreckels et al. have investigated colonization of  Lactobacillus reuteri DSM 17938 in preterm ELBW infants from birth to postmenstrual week (PMW) 36 in a randomized manner and looked at body composition, health metrics. They not only found successful colonization of L. reuteri but also this colonization improved head growth from birth to four weeks of age. 

Response to reviewer 2

We thank the reviewer for revising our manuscript, the comments, and questions. Below, we answer each comment individually and hope that our answers clarify the decisions we made.

Why did the authors specifically choose L. reuteri and ? There are multiple strains of probiotics in market, in testing phases and isolation. Each strain works in a individual way. The probiotic benefits may depend on it survival of gastric environment and reaching in enough numbers. 

  1. reuteri DSM 17938 was selected because this strain improved feeding tolerance, reduced sepsis, and shortened the length of the hospital stay in a previous randomised controlled trial including very low birth weight (VLBW, <1,500 g) and extremely low birth weight (ELBW, <1,000 g) infants born before gestational week (GW) 32 (Oncel et al. Arch. Dis. Child. Fetal Neonatal Ed. 2014). Additionally, a retrospective cohort study reported a significant reduction of necrotising enterocolitis (NEC) and a trend towards less sepsis and lowered mortality in ELBW infants (Hunter et al. BMC Pediatrics. 2012). With our trial, we aimed to investigate if these beneficial effects were present in the even more vulnerable extremely preterm (birth before GW 28) ELBW infant group using a randomised placebo-controlled trial study design.

After the start of the PROPEL trial, two additional studies reported beneficial effects of L. reuteri on low birth weight (LBW) infants: First, Cui et al (2019) reported that L. reuteri DSM 17938 supplementation shortened the time to full enteral feeding and the duration of hospitalisation of infants weighing between 1,500 and 2,000 g (Cui et al. Ital. J. Pediatr. 2019). Second, Kaban et al (2019) described that L. reuteri DSM 17938 supplementation reduced feeding intolerance and tended to reduce NEC and to lower mortality in infants weighing between 1,000 and 1,800 g (Kaban et al. Pediatr. Gastroenterol. Hepatol. Nutr. 2019). In addition, a strain-specific meta-analysis published in 2018 showed a reduction of NEC and a shorter time to full enteral feeding in preterm infants (birth before GW 37) who received L. reuteri DSM 17938 (van den Akker et al. J. Pediatr. Gastroenterol. Nutr.2018). Importantly, this meta-analysis also pointed out that studies in the ELBW infant population are scarce.

We have now extended the introduction of the manuscript to include more information about our rationale for choosing L. reuteriDSM 17938:

Introduction, lines 38-55: “Meta-analyses suggest that probiotics reduce the incidence of necrotising enterocolitis (NEC), death, and NEC-related death in very low birth weight (VLBW, <1,500 g) infants [2,3], but also that effects are strain-dependent [4]. One promising candidate strain is Lactobacillus reuteri DSM 17938. Three randomised controlled trials (RCT) report beneficial effects of this strain for low birth weight infants. First, L. reuteri DSM 17938 was shown to improve feeding tolerance, reduce sepsis, and shorten the duration of hospitalisation of VLBW and ELBW infants born before GW 32 [5]. Second, supplementation of this L. reuteri strain shortened the time to full enteral feeding and the duration of hospitalisation of infants with a birth weight between 1,500 and 2,000 g [6]. Third, it also reduced feeding intolerance and tended to reduce NEC and to lower mortality in infants weighing between 1,000 and 1,800 g at birth [7]. In addition, a meta-analysis showed a reduction of NEC and a shorter time to full enteral feeding in preterm infants (born before GW 37) supplemented with L. reuteri DSM 17938 [4]. Importantly, this meta-analysis also pointed out that studies in the ELBW infant population are scarce. Only one of the three mentioned RCTs investigated the benefits of L. reuteri DSM 17938 specifically in ELBW infants [5]. The efficacy of probiotics such as L. reuteri DSM 17938 in the subgroup of ELBW infants thus remains questioned [4].”

Furthermore, for L. reuteri ATCC 55730, the parent strain of L. reuteri DSM 17938, published studies already showed that the strain could colonise stomach, duodenum, and ileum of healthy adults (Valeur et al. Appl. Environ. Microbiol. 2004), and that it was viable in full-term infants’ intestines (Abrahamsson et al. J. Pediatr. Gastroenterol. Nutr. 2009). Since we agree with the reviewer that the survival of probiotics might be an important factor for allowing the bacteria to exert beneficial effects, this previous data further encouraged us to test whether L. reuteri DSM 17938 could colonise and be beneficial for extremely preterm ELBW infants. We have also cited those studies on L. reuteri colonisation in the introduction of our manuscript:

Introduction, lines 66-68: “The mother strain of L. reuteri DSM 17938, L. reuteri ATCC 55730, can colonise stomach, duodenum, and ileum of healthy adults [12], and viable L. reuteri are present in infant faeces after supplementation [13].”

The authors in their previous research (reference# 6) have shown oral supplementation with oil drops. Did the authors test if the oil have any effect on probiotic efficacy? is synergistic effect ? Where any stability studies conducted ? 

The product was developed as oil drops to facilitate the supplementation of the probiotic and placebo products to infants, including those infants who require feeding via a nasogastric tube. Since the placebo was provided in an identical oil suspension as the probiotic product, our trial investigated the effect of L. reuteri vs. placebo (maltodextrin), and the oil drop formulation should not have affected between-groups and within-group comparisons. The manufacturer, BioGaia AB, regularly controlled the quality of the study products and the concentration of L. reuteri was within the stipulated limits in all batches that were used in the PROPEL trial.

We have not performed trials to compare different vehicles and whether they affected L. reuteri efficacy. However, other studies, including the ones specifically on ELBW infants mentioned above (Oncel et al. Arch. Dis. Child. Fetal Neonatal Ed. 2014, Hunter et al. BMC Pediatrics. 2012), also used the oil drop-based formulations from BioGaia AB and found L. reuteri to exert beneficial effects. In line with these studies, our trial indicates that L. reuteri can colonise the extremely preterm ELBW infants when supplied in oil drops.

It would have been beneficial to compare with a standard or already available L. acidophilus or Bifido strains? 

We agree that it is interesting to compare different probiotic products within one trial. This was, however, not the scope of the PROPEL trial. With the PROPEL trial, we aimed to investigate whether supplementation of L. reuteri DSM 17938 to the most premature (GW <28) ELBW infants was safe and whether beneficial effects could be observed in this most vulnerable population in Swedish neonatal intensive care units as well. Comparison of several probiotic products (and their respective placebo products) within one trial would have required a much larger sample size and, luckily, the percentage of ELBW infants in Sweden is very small (<1%), so that the absolute number of infants who could potentially be recruited for a trial is limited. We thus decided to focus on one promising candidate, L. reuteri DSM 17938, instead of comparing several probiotic products in one trial.

To compare the results of our trial to studies using different probiotics, we refer to randomised controlled trials on preterm infants using other Lactobacillus or Bifidobacterium strains in the discussion section of this manuscript:

Discussion, lines 443-450: “A study by Rougé et al. (2009) [10] on ELBW infants and the PiPS study [11] on very preterm infants suggest that the colonisation with and effects of the supplemented strains Bifidobacterium longum BB536 and L. rhamnosus GG or B. breve BBG-001, respectively, were hindered by antibiotic treatment. Their findings are in contrast to the general results from our study, which might be explained by different antibiotic resistance patterns of the probiotic strains. Unfortunately, data on different antibiotic classes was not presented in the two studies, so it remains unclear which antibiotic class(es) hindered probiotic growth in these previous studies.”

Discussion, lines 459-468: “In contrast to the PiPS trial [11], low gestational age did not negatively affect colonisation in this study. L. reuteri colonisation at one week of age was dependent on the number of hours that an infant received the probiotic product between birth and faecal sampling at one week, and L. reuteri prevalence at PMW 36 was lower if infants had more days between birth and PMW 36 without the probiotic product. These findings highlight that L. reuteri colonisation depends on regular L. reuteri intake, which is in agreement with Rougé et al. [10], who describe that suspension of enteral feeding and the associated suspension of probiotic administration possibly lead to a lack of colonisation of ELBW infants with B. longum BB536 and L. rhamnosus GG.”

Discussion, lines 476-478: “The lack of effect of feeding mode on L. reuteri colonisation in our RCT is also in contrast to the PiPS trial [11], in which breast milk reduced colonisation of very preterm infants with probiotic B. breve BBG-001.”

Discussion,lines 497-498, 516-523: “Besides poor colonisation of the probiotic group, cross-colonisation of the placebo group with the probiotic bacteria has been suggested as an explanation for a lack of effects of probiotics in preterm infants [11]. In our study, probiotic L. reuteri were detected in a maximum of 10% of the faecal samples of the placebo group. […] In line with our recent report on low L. reuteriprevalence and abundance in placebo-supplemented infants [14], this suggests that it is unlikely that cross-colonisation of placebo-supplemented infants causes the lack of effects of L. reuteri supplementation in our trial.”

What is the source of isolation of the probiotic strain?

The parent strain of the probiotic strain used in the PROPEL trial, L. reuteri ATCC 55730, has been isolated from breast milk of a Peruvian mother (Srinivasan et al. Pediatr. Ther. 2018). Removal of resistance gene-carrying plasmids from the parent strain generated the L. reuteri DSM 17938 strain (Rosander et al. Appl. Environ. Microbiol. 2008), which is used in this study.

We have now added this information to the Materials and Methods section:

Materials and Methods, lines 96-117 “L. reuteri DSM 17938 has been generated by removal of two resistance gene-carrying plasmids from its mother strain L. reuteri ATCC 55730 [15], which was originally isolated from the breast milk of a Peruvian mother [16].

Figure 2, colonization of L. reuteri, the authors showed the presence of L. reuteri from 2nd weeks placebo groups as well, but not first week, was there any correlation and influence of diet & feeding?

All extremely preterm ELBW infants in our trial were exclusively fed breast milk until a weight of at least 2,000 kg, as mentioned in the methods section of the manuscript:

Materials and Methods, lines 130-131: “All infants were exclusively fed mother’s own or donor milk until they reached a weight of at least 2,000 g.”

Hence, we were not able to investigate the effect of differences in infant feeding mode on early colonization with the probiotic L. reuteri. We could, however, investigate the effect of infant feeding type on colonization at PMW 36 since the feeding type could differ between infants at this time point. Comparison of formula-fed, partially breast milk-fed, and exclusively breast milk-fed infants showed no significant differences on L. reuteri prevalence and abundance (Figure 5). We have mentioned this in the results section:

Results,lines 311-312, 325-328: “Furthermore, we studied whether infant feeding influenced L. reuteri colonisation. All infants were exclusively fed mother’s own or donor milk until they weighed at least 2,000 g, thus investigating the effect of feeding mode on early L. reuteri colonisation was not possible. There was no significant difference in L. reuteri prevalence or abundance in faeces of L. reuteri-supplemented infants at PMW 36 when comparing formula-fed, partially breast milk-fed, and exclusively breast milk-fed infants (Figure 5).”

Looking at microbiome and showing the changes in the microbial communities specifically different Lactobacillus species in the gut w/ or w/o probiotic supplementation would be beneficial. 

 The gut microbiota of the infants from the PROPEL trial was analysed using 16S rRNA gene sequencing, and we recently published a study comparing the gut microbiota of the placebo and L. reuteri-supplemented infants (Martí et al. Cell Reports Med.2021). Briefly, supplementation with L. reuteri associated with an increased gut bacterial diversity at one to four weeks of age, and with higher L. reuteri abundance at two to four weeks of age. Additionally, L. reuteri supplementation was related to a lower abundance of potentially pathogenic Enterobacteriaceae and Staphylococcaceae at one week of life. Unfortunately, the 16S rRNA gene sequencing approach did not allow us to compare different Lactobacilli species since only a small number of reads could be assigned at species level. Hence, comparison between different Lactobacilli species was not possible.

Reviewer 3 Report

[General comments]

It is now common problem that preterm infants, born either naturally or through caesarean section, will suffer from neurological disorders in later life. Therefore, to breed preterm infants in health is highly important. This study was conducted as a part of The PROPEL trial where the effect of Lactobacillus reuteri on the growth and clinical condition in preterm infants was analyzed. The results are informative and indicate the points to care when probiotics will be used for keeping health of preterm infants.

[Specific comments]

  1. Even if the same dose of reuteri was administered, about 14% of infants did not colonize this bacterium. Through the correlation analyses, the authors point out that the interval of administration and oligosaccharide composition in breast milk may be important factors. Are the microbiota composition between colonizers and non-colonizers in L. reuteri-supplemented group before administration, different or comparable?
  2. The finding that reuteri abundance at one week of age is associated with sepsis (Line 332-334), is important in terms of probiotics safety. Which mechanisms can be suggested as an influence of administered L. reuteri?
  3. It is interesting that the head growth has been improved by giving reuteri in preterm infants (Figure 7). Is this phenomenon due to the improvement of nutritional absorption and/or intake?
  4. To assess the effect of reuteri administration on clinical outcomes, non-colonized placebo group and L. reuteri-colonized L. reuteri group was compared. When the all placebo infants and all L. reuteri-administered infants (including both colonizers and non-colonizers), is the conclusion different?

Author Response

Comments from reviewer 3

It is now common problem that preterm infants, born either naturally or through caesarean section, will suffer from neurological disorders in later life. Therefore, to breed preterm infants in health is highly important. This study was conducted as a part of The PROPEL trial where the effect of Lactobacillus reuteri on the growth and clinical condition in preterm infants was analyzed. The results are informative and indicate the points to care when probiotics will be used for keeping health of preterm infants.

Response to reviewer 3

We thank the reviewer for the positive feedback and additional comments. Below, we answer each specific comment separately and hope that our answers will satisfy the reviewer.

  1. Even if the same dose of reuteri was administered, about 14% of infants did not colonize this bacterium. Through the correlation analyses, the authors point out that the interval of administration and oligosaccharide composition in breast milk may be important factors. Are the microbiota composition between colonizers and non-colonizers in L. reuteri-supplemented group before administration, different or comparable?

We indeed show that at least 86% of the L. reuteri-supplemented infants were colonised with the probiotic L. reuteri strain. However, this does not mean that 14% of the L. reuteri-supplemented infants were never colonised with L. reuteri. The L. reuteri prevalence of 86% was found for infants at one week of age, while at two, three, and four weeks of age, over 90% of the infants were colonised with L. reuteri (see Figure 2a). Additionally, all L. reuteri-supplemented infants had a faecal sample positive for the probiotic L. reuteri strain at least once, with the exception of one infant, which passed away before postmenstrual week (PMW) 36 and for which faeces was only collected at one week of age. We have now added the latter information to the results section:

Results, lines 240-242: “All L. reuteri-supplemented infants had a faecal sample positive for the probiotic L. reuteri strain at least once, with the exception of one infant, which passed away before PMW 36 and for which faeces was only collected at one week of age.”

We apologise that our presentation was misleading and hope that we could clarify that our data does not support the statement that about 14% of the infants were not colonised with L. reuteri, but rather that, in some infants, L. reuteri was not detected in infant faeces early after starting probiotic supplementation and when supplementation was paused. We mention these findings in the results and discussion sections of our manuscript:

Results, lines 272-276: “The number of hours with the probiotic product before faecal sample collection at one week of age was significantly higher in L. reuteri-colonised compared to non-colonised infants (Table 1a). Similarly, L. reuteri abundance at one week of age was positively correlated with the number of hours with the probiotic product (Spearman’s rho = 0.55, Figure S1g).”

Results, lines 288-290: “The number of days without the probiotic product before PMW 36 was significantly higher in infants who were not colonised compared to infants who were colonised with L. reuteri at PMW 36 (Table 1b).”

Discussion, lines 460-468: “L. reuteri colonisation at one week of age was dependent on the number of hours that an infant received the probiotic product between birth and faecal sampling at one week, and L. reuteri prevalence at PMW 36 was lower if infants had more days between birth and PMW 36 without the probiotic product. These findings highlight that L. reuteri colonisation depends on regular L. reuteriintake, which is in agreement with Rougé et al. [10], who describe that suspension of enteral feeding and the associated suspension of probiotic administration possibly lead to a lack of colonisation of ELBW infants with B. longum BB536 and L. rhamnosus GG.”

Unfortunately, no faecal samples were collected before the start of the supplementation, so we are not able to investigate how the initial gut microbiota composition affected L. reuteri colonisation.

We recently published a paper on the gut microbiota composition of the infants in the PROPEL trial (Martí et al. Cell Reports Med. 2021). In the paper, we report that the gut microbiota composition differs between the placebo and L. reuteri-supplemented group during the first four weeks of life. In addition, for the second, third, and fourth week of life, L. reuteri DSM 17938 abundance associated with the gut microbiota composition of the L. reuteri-supplemented group (see Figure 3 in Martí et al. Cell Reports Med. 2021).

Using the same method as in our recent publication, non-metric multidimensional scaling (NMDS), we now investigated the similarity of the gut microbiota composition of non-colonized and L. reuteri-colonised infants from the placebo and L. reuteri-supplemented group (see Figure R1). In Table R1, we additionally show the number of samples for the different groups shown in Figure R1. The NMDS plots show that L. reuteri colonisation determines the gut microbiota composition at one, two, three, and four weeks of age. Although the number of non-colonized infants from the L. reuteri-supplemented group is small, the non-colonized infants tend to cluster with non-colonized infants from the placebo group, especially at two and three weeks of age. Vice versa, the L. reuteri-colonised infants from the placebo group tend to cluster with the L. reuteri-colonised infants from the L. reuteri-supplemented group.

Figure R1: Non-metric multidimensional scaling (NMDS) of bacterial community composition at one, two, three, and four weeks, at PMW 36, and at 2 years of age. Colours indicate the supplementation group (yellow = placebo, blue = L. reuteri) and the filling of the dots depicts whether an infant was colonised with the probiotic L. reuteri strain at the respective time point (empty dot = non-colonised, filled dot = L. reuteri-colonised).

Table R1: Number of infants in the respective groups per time point. Note that the number of infants can be lower than in the manuscript since gut microbiota data was not available for all samples included in this manuscript.

Placebo group

L. reuteri group

Time point

Non-colonised

L. reuteri-colonised

Non-colonised

L. reuteri-colonised

1 week

54

0

8

46

2 weeks

53

2

2

52

3 weeks

46

5

4

47

4 weeks

43

5

1

52

PMW 36

39

2

7

43

2 years

26

1

20

0

  1. The finding that reuteri abundance at one week of age is associated with sepsis (Line 332-334), is important in terms of probiotics safety. Which mechanisms can be suggested as an influence of administered L. reuteri?

We agree that the finding of increased sepsis prevalence in L. reuteri-supplemented infants with higher L. reuteri abundance at one week of age is important when considering whether the probiotic administration is safe for the vulnerable extremely preterm extremely low birth weight (ELBW) infants. However, we would also like to point out that this association between L. reuteri abundance and sepsis was only detected when using L. reuteri abundance data at one week of age and not when using data from two, three, and four weeks of age, as mentioned in the results section:

Results, lines 391-394: “L. reuteri abundance in faeces at one week of age was also significantly higher in infants who later developed culture-proven sepsis (Figure 8c). Of note, L. reuteri abundance in faeces at two, three, and four weeks of age was not associated with a higher risk of culture-proven sepsis.”

As described in the discussion section, we believe that the finding of a potential increase in sepsis incidence should be treated with caution, but we also agree that clinicians and researchers should remain cautious with the administration of the probiotic L. reuteristrain to the extremely preterm ELBW infants:

Discussion, lines 529-546: “However, when investigating the effects of L. reuteri abundance on clinical outcomes in the L. reuteri-supplemented group, we also detected a potential increase in culture-proven sepsis in infants, who had a higher L. reuteri abundance in faeces at one week of age. This effect was seen neither in the intention-to-treat analysis in the original publication on the clinical outcomes of these extremely preterm ELBW infants [8] nor when comparing non-colonised placebo and L. reuteri-colonised L. reuteri-supplemented ELBW infants in the current study. Importantly, there were no cases of sepsis with the probiotic L. reuteri. Our RCT was not powered to detect differences in sepsis incidence, and the analysis of sepsis occurrence was based on a subset of only 53 ELBW infants. L. reuterisupplementation has further been described to reduce sepsis in a previous RCT in 196 ELBW infants [5] using a similar dose as in our trial (1 x 108 vs. 1.25 x 108 bacteria per day in our RCT). Although our results should be interpreted with caution, a potentially shorter time to reach full enteral feeding and a potentially increased sepsis incidence with higher L. reuteri abundance indicate that changing treatment doses in extremely preterm ELBW infants could have both positive and negative effects on clinical outcomes. In a previous trial, infants with a birth weight below 750 g were more likely to suffer from sepsis than infants with a birth weight between 750 and 1,500 g [38], and it is thus important to remain cautious with the administration of probiotics to the most vulnerable extremely preterm ELBW infants.”

The mechanism that could underlie a potential increase in sepsis incidence after L. reuteri administration remains unclear. The functions of L. reuteri DSM 17938 have been studied intensively and include production of anti-microbial agents and short chain fatty acids, inhibition of intestinal inflammation, and altered gut motility (Srinivasan et al. Pediatr. Ther. 2018). In line with its generally anti-inflammatory, health-promoting functions, L. reuteri seems to reduce necrotising enterocolitis and late-onset sepsis in certain preterm infant populations (Srinivasan et al. Pediatr. Ther. 2018).

Since L. reuteri can reduce intestinal inflammation and hereby helps to maintain the integrity of the intestinal barrier, it is unexpected that L. reuteri might facilitate the translocation of other microbes into the blood stream and hereby increases sepsis occurrence. Additionally, in our recent paper reporting on the gut microbiota composition of the infants in the PROPEL trial, we describe that the relative abundance of potentially pathogenic Enterobacteriaceae and Staphylococcaceae were lower in faeces at one week of age in the L. reuteri-supplemented compared to the placebo group (Martí et al. Cell Reports Med. 2021).

Nevertheless, one could speculate that the anti-inflammatory functions of L. reuteri might allow other pathogens to grow and cross the epithelial barrier without immune recognition. The relative abundance of Clostridia in the intestines of the extremely preterm infants in our trial was higher in infants experiencing sepsis compared to infants not experiencing sepsis later (Martí et al. Cell Reports Med. 2021). Possibly, pathogenic Clostridia were thus able to thrive in the infants’ intestines, translocate into the infants’ blood, and lead to sepsis. However, this hypothesis remains speculative and would need experimental support.

  1. It is interesting that the head growth has been improved by giving reuteri in preterm infants (Figure 7). Is this phenomenon due to the improvement of nutritional absorption and/or intake?

We agree that it is an interesting finding that L. reuteri improves head growth of extremely preterm ELBW infants. This finding should, however, be interpreted with caution since our trial was not powered to detect differences in growth rates between the supplementation or colonisation groups. Akar et al (Akar et al. J. Matern. Neonatal Med. 2017) found no effect of L. reuteri supplementation on neuromotor, neurosensory, and cognitive outcomes of very low birth weight (VLBW) infants at 18 to 24 months corrected age. For the infants from our trial, long-term neurological outcome assessment remains to be done.

Published studies show that the gut microbiota is linked to nutrition and affects infant growth (Blanton et al. Science. 2016, Chen et al. N. Engl. J. Med.  2021), and that probiotic supplementation potentially improves weight and length growth of (undernourished) infants in developing countries (Onubi et al. J of Health, Population and Nutr. 2015). L. reuteri DSM 17938 supplementation improves infant weight and height growth in one- to six-year-old Indonesian infants (Agustina et al. J. Nutr.2013), but data specifically on infant head growth is not available.

In our study, we found that L. reuteri colonisation and higher L. reuteri abundance shorten the time to full enteral feeding within the group of L. reuteri-supplemented infants (Figure 8). We could thus speculate that L. reuteri might have improved the infants’ nutrition and hereby positively affected their growth. The gut microbiota can influence the peripheral and central nervous system and host metabolism via production of neurotransmitters and short chain fatty acids (Bienenstock et al. Nutr. Rev. 2015, O’Mahony et al. Behav. Brain. Res. 2015). A study in undernourished Bangladeshi children at 12 to 18 months of age describes changes in concentrations of blood plasma proteins linked to bone growth and neurodevelopment due to manipulation of the gut microbiota (Chen et al. N. Engl. J. Med.  2021). Different Lactobacilli, including L. reuteri (Romeo et al. J. Perinatol. 2010, Buffington et al. Cell. 2016), positively affect neurological outcomes of preterm infants or mice pups (Lebovitz et al. Brain Behav. Immun. 2019, Patterson et al. Sci. Rep. 2019, Lu et al. Sci. Rep. 2020). However, data from human, especially infant, studies is still scarce and, unfortunately, the study design of the PROPEL trial did not allow us to further investigate mechanisms behind a potentially improved nutrient uptake and growth. Future trials including metabolite analyses and functional in vitro and in vivotests are needed to better understand how the gut microbiota, host metabolism, and brain development and function are linked.

In the discussion section of our manuscript, we have now included some of this information on how the gut microbiota and neurological outcomes could be linked:

Discussion, lines 547-566: “In line with the first paper about this trial [8], we describe here an improved head growth in female and L. reuteri-colonised extremely preterm ELBW infants. Since L. reuteri colonisation and higher L. reuteri abundance shorten the time to full enteral feeding within the L. reuteri-supplemented infant group, it is tempting to speculate that L. reuteri improved the infants’ nutrition and hereby positively affected their growth. The gut microbiota can influence the peripheral and central nervous system and host metabolism via production of neurotransmitters and short chain fatty acids [39,40]. A study in undernourished Bangladeshi infants at 12 to 18 months of age describes changes in concentrations of blood plasma proteins linked to neurodevelopment due to manipulation of the gut microbiota [41]. Although the head growth of the infants in our trial is still inferior to the growth in utero, an increased head circumference growth rate might be associated with improved neurodevelopment of extremely preterm infants [42]. Intake of L. reuteri during the neonatal period might thus prevent or ameliorate neurological disabilities frequently present in premature infants [43]. Indeed, supplementation of L. reuteri ATCC 55730, the mother strain of L. reuteri DSM 17938, positively affected neurological outcomes of preterm infants born before GW 37 with a birth weight below 2,500 g [44]. However, another study reports no effect of L. reuteri DSM 17938 on neuromotor, neurosensory, and cognitive outcomes of VLBW infants at 18 to 24 months corrected age [45]. To ascertain whether the improved head growth detected in this study is of clinical importance, neurological outcome assessment of study participants at an older age is planned.”

  1. To assess the effect of reuteri administration on clinical outcomes, non-colonized placebo group and L. reuteri-colonized L. reuteri group was compared. When the all placebo infants and all L. reuteri-administered infants (including both colonizers and non-colonizers), is the conclusion different ?

Our research group previously reported the results of L. reuteri vs. placebo supplementation on feeding intolerance, clinical outcomes, and growth in the PROPEL trial (reference 8 in the manuscript, Wejryd et al. Acta Pediatr. 2019). L. reuterisupplementation affected neither feeding intolerance nor morbidities, but improved head growth from birth to four weeks of age, compared to placebo supplementation. Additionally, female gender was associated with improved head growth from birth to four weeks of age. We have briefly mentioned the main results from this study in the introduction of this manuscript:

Introduction, lines 58-60: “We did not find any effect on feeding intolerance, NEC, and sepsis, but head growth improved in the L. reuteri-supplemented group [8]”

The beneficial effect of L. reuteri on head growth was found both when comparing infants by supplementation group (placebo vs. L. reuteri group, Figure 2 from Wejryd et al. Acta Pediatr. 2019) and when comparing non-colonised infants from the placebo group with L. reuteri-colonised infants from the L. reuteri group (Figure 7 of this manuscript). The described lack of effect of L. reuteri on feeding intolerance and clinical outcomes like necrotising enterocolitis (NEC) and sepsis was also found both when comparing infants by supplementation group (Tables 2 and 3 from Wejryd et al. Acta Pediatr. 2019) and by L. reuteri colonisation (Figure 7 of this manuscript). Thus, the conclusions from both comparisons are the same, and we have mentioned this in the discussion section of this manuscript:

Discussion, lines 517-520: “Moreover, comparison of clinical outcomes between L. reuteri-colonised L. reuteri-supplemented and non-colonised placebo-supplemented infants confirms the lack of effect described in the original trial that compared L. reuteri and placebo supplementation groups [8].”

Discussion, lines 547-548: “In line with the first paper about this trial [8], we describe here an improved head growth in female and L. reuteri-colonised extremely preterm ELBW infants.”

On top of between-supplementation-group comparisons, the quantitative PCR data in this manuscript additionally allowed us to investigate outcomes within the group of L. reuteri-supplemented infants. Within the L. reuteri-supplemented infant group, we describe that a higher L. reuteri prevalence and abundance at one week of age shorten the time to full enteral feeding and that a higher L. reuteri abundance in faeces at one week of age is associated with more sepsis cases later (Figure 8 of this manuscript).
